# Uterine Fibroids and Pregnancy: A Review of the Challenges from a Romanian Tertiary Level Institution

**DOI:** 10.3390/healthcare10050855

**Published:** 2022-05-06

**Authors:** Mihaela Camelia Tîrnovanu, Ludmila Lozneanu, Ştefan Dragoş Tîrnovanu, Vlad Gabriel Tîrnovanu, Mircea Onofriescu, Carmen Ungureanu, Bogdan Florin Toma, Elena Cojocaru

**Affiliations:** 1Department of Mother and Child Medicine, “Grigore. T. Popa” University of Medicine and Pharmacy, 700115 Iasi, Romania; mihaela.tirnovanu@umfiasi.ro (M.C.T.); vlad.tirno@gmail.com (V.G.T.); mirceaonofriescu@yahoo.com (M.O.); bogdan-florin-sl-toma@d.umfiasi.ro (B.F.T.); 21st Clinic of Obstetric and Gynecology, Cuza Voda Obstetrics-Gynecology Clinic Hospital, 700038 Iaşi, Romania; 3Department of Morphofunctional Sciences I, “Grigore T. Popa” University of Medicine and Pharmacy, 700115 Iasi, Romania; elena2.cojocaru@umfiasi.ro; 4Department of Surgery II, “Grigore T. Popa” University of Medicine and Pharmacy, 700115 Iasi, Romania; stefan-dragos.tirnovanu@d.umfiasi.ro

**Keywords:** pregnancy, uterine fibroid, myomectomy, ultrasound, obstetric outcome

## Abstract

Background and Objectives. Uterine fibroids are relatively common in reproductive-age women and are directly linked to pregnancy. There are many debates about performing a myomectomy at the same time as a caesarian section (CS) in such cases due to the risk of a hemorrhage. Our study aims to investigate fibroid features and their evolution in pregnancy, complications of a myomectomy during CS and maternal and fetal obstetric outcomes of pregnancies with fibroids. Materials and methods. We realize a prospective study that includes 57 patients with fibroids in pregnancy diagnosed in January 2017–June 2019. We analyze the number, the location and the growth of fibroids during pregnancy and the maternal and fetal outcome. We appreciate the bleeding in patients with a myomectomy and without a myomectomy during CS, using hemoglobin values before and after birth. Results. Most of the patients present single fibroids that are 30–160 mm in size, located on the anterior uterine wall. Vaginal delivery is used in 7% of women, whereas 85.96% deliver by CS. In addition, 68% of fibroids are diagnosed in the first trimester. In most cases, the fibroid has maximum growth in the second trimester of pregnancy. The myomectomy rate for fibroids during CS is 24.48. Hemoglobin values showed no statistically significant difference between the two groups with and without myomectomy. The operating time is double for the group with a myomectomy associated with a CS. The results of the obstetric outcomes are abortion in 7% of all patients, whereas premature delivery and births at term are 9.43% and 90.57%, respectively. Conclusions. The decision of performing a myomectomy during pregnancy can be a challenge and must be performed for selected cases. This procedure may have several benefits, such as avoiding another operation to remove fibroids.

## 1. Introduction

Uterine fibroids or myomas represent benign hormone-dependent tumors of smooth muscle on the uterine wall, occurring in 20–60% of women of reproductive age [1]. The development of fibroids is related to various risk factors, such as age, race, hormonal factors, uterine infection, obesity and behavioral factors, but the epidemiological data are inconsistent. Generally, they are asymptomatic, and only about 30% of these fibroids are large enough to be discovered by a health care practitioner during a physical examination [2]. In addition, approximately one-third of them develop serious clinical manifestations, such as abnormal uterine bleeding, anemia, back pain, pelvic pain and pressure, constipation, urinary frequency or infertility, thus necessitating treatment. Moreover, fibroids have been linked to poor obstetric outcomes [3]. 

Fibroids are found in 0.1–10.7% of pregnant women, and their prevalence rises if women want to postpone having children until a later age [4]. Pregnancy-related hormones influence the size of uterine fibroids, and fibroids have many impacts on pregnancy. Women with uterine fibroids in pregnancy generally have concerns related to adverse outcomes. However, these women generally have uneventful outcomes in pregnancy. Several studies have reported inconsistent relationships between uterine fibroids and adverse obstetric outcomes. Miscarriage, premature labor, antepartum hemorrhaging, malposition, malpresentation, obstructed labor, uterine inversion, postpartum hemorrhaging and puerperal sepsis are among the obstetric consequences of co-existing uterine fibroids in pregnancy [5].

It is known that uterine fibroids, especially multiple, intramural or submucous, are associated with an increased risk of early pregnancy loss [6,7]. Fibroids developed in the uterine body are more likely to cause miscarriage than fibroids in the lower uterine area. Enhanced uterine irritability and contractility are suggested factors that lead to increased pregnancy loss when there is a co-existing fibroid [8,9], a compressive effect and a disruption of blood flow to the placenta and fetus. This is more likely when the placenta implants close to a fibroid nodule, and this depends on the location of the uterine fibroids [9]. 

Several studies have shown that women with uterine fibroids have a greater risk of imminent preterm labor and preterm births in late pregnancies [5,10,11,12]. Epidemiological data prove that uterine myomas do not appear to increase the incidence of preterm premature ruptures of membranes, although other studies have found the opposite [13,14].

The goal of our study is to analyze the prevalence of fibroids, their features (number, size, location, type), complications of a myomectomy during a caesarian section (CS) and maternal and fetal prognoses among pregnant women admitted to the Iasi ‘Cuza Voda’ University Hospital of Obstetrics and Gynecology in Romania, a tertiary level hospital.

## 2. Materials and Methods

In our prospective study, we analyzed 57 pregnant women hospitalized at our clinic with uterine fibroids, diagnosed both antenatally and during surgery. The patients were selected from a total number of 7600 births in 2017–2019. A total of 49 of the 57 cases gave birth by CS, and these were divided into two groups: 12 women who had a CS with a myomectomy and 37 without a myomectomy. 

All the women from the study had regular prenatal consultation from the first trimester of pregnancy. Fibroids were assessed by ultrasound (2D examination) during each trimester of pregnancy. They were measured in two perpendicular planes, twice. During examination, we looked at the number of fibroids, their size, location and type. In addition, we analyzed comparative blood loss between cases with and without a myomectomy using hemoglobin as an assessment parameter before and 2 days after CS, as well as blood transfusion necessity, postoperative temperature, the duration of operation and the length of hospital stay. The study was conducted according to the guidelines of the Declaration of Helsinki and approved by the Research Ethics Committee of the Cuza Vodă Obstetrics and Gynecology University Hospital in Iasi.

For statistical analysis we used Kruskal–Wallis ANOVA, Student’s *t*-test and a chi-squared test. A *p* value of <0.05 was considered significant. 

## 3. Results

Our patients were selected from a total of 7600 births in 2017–2019 as follows: 3064 in 2017 (40.31%), 2888 in 2018 (38%) and 1648 in 2019 (21.69%). We finally enrolled in the study 57 (0.75%) pregnant women admitted in our clinic with diagnoses of uterine fibroids documented antenatally and at surgery. The distribution of cases by year shows the following prevalence: 42.1% (24 cases) in 2017, 33.33% (19 cases) in 2018 and 24.56% (14 cases) in 2019. From the total number, 4 patients had normal births, 4 had abortions and 49 had caesarean sections (Table 1).

As shown in the table, 68% of cases were diagnosed in the first trimester, 29% were diagnosed in the second trimester and 3% were diagnosed in the third trimester of pregnancy. The ages range between 25 and 45 years. Most cases (23 cases) are in the 30–34 age group (40.35%), followed by those with ages between 35 and 39 years (20 cases) (35.08%). There were also 5 women (8.77%) under the age of 30, and 9 women (15.78%) were over the age of 40. The first delivery was registered in 39 cases (68.42%). In our study, only 11 pregnant women were obese (Table 1).

The proportion of women with fibroids diagnosed during pregnancy (31 cases) is slightly higher than that of cases with fibroids diagnosed before pregnancy (26 cases) (54.4% vs. 45.6%). For 9 cases, the fibroids were diagnosed at birth by CS (Figure 1).

In our study, single fibroids occurred more frequently, with 42 cases (74%) compared to those with multiple fibroids (26%) (Figure 2). Women with two fibroids represent 6 cases (11%), with ages varying between 32 and 40 years old. Those with three fibroids represent 5 cases (9%), with ages varying between 31 and 40 years old. Women with five fibroids represent 2 cases, with ages varying between 38 and 41 years old. Finally, women with seven fibroids represent 2 cases (4%), with ages varying between 31 and 40 years old. 

The two most frequent types of myoma were interstitial and subserous, having 28 cases (49.12%) and 26 cases (45.61%), respectively, whereas 3 cases (5.26%) were submucous. The location of the myoma was on the anterior uterine wall in 50.87% of cases, followed by the fundic region in 19.3% of cases. In addition, 12.28% developed on the posterior wall (7 cases), and they were on the isthmic region (6 cases) and the right (1 case) and left (3 cases) lateral walls (Table 1). The average size of the myoma was 6.2 cm with limits between 30 mm and 160 mm.

In most cases, the increase of fibroids was continuous (Figure 3 and Figure 4), except for 5 cases in which they decreased compared to their sizes in the second trimester. In only 1 case, there was a decrease in the third trimester compared to first trimester, whereas in some cases, fibroids remained constant in size during pregnancy. As the results of our study show, in most cases, the maximum increase of uterine fibroids occurred in the second trimester (Table 2) and not in the first trimester under the influence of estrogen and HCG, as we expected (Figure 5a–c). 

A total of 26 patients in the study were diagnosed with fibroids before pregnancy, so we can assess their growth during the first trimester as follows: 10 patients had stationary dimensions (38.46%), 3 cases increased by 25%, 4 cases increased by 50%, 2 cases increased by 75%, 4 cases increased by 100%, 2 cases increased by 150% and 1 case increased by 200%. 

Occasionally, we have problems with the choice of the site of puncture during amniocentesis if the fibroid has an anterior location (Figure 6 and Figure 7), especially if the placenta also has an anterior position.

Regarding the evolution of the pregnancy for our 57 patients, we had 4 abortions (7%), 5 premature births (9.43%), 8 cases with treatment for imminent premature birth (14%) and 8 for imminent abortion (14%), 9 patients had a premature rupture of membranes (16%) and in 1 case (2%) there was an abruptio placentae. In cases with abortions, 3 pregnancies stopped in evolution.

When we discuss the influences of fibroids on the fetus, 4 cases had fetal dystocia by breach presentation, 4 cases had intrauterine growth restriction and 2 babies were small for their gestational age. The statuses of the fetuses were very good at delivery, with only 1 having an Apgar score of 4.

In our study, 4 cases (7%) gave birth by vaginal delivery, and 49 gave birth after CS (85.96%). Regarding the indications for those who had a cesarean section, 21 (42.86%) cases presented praevia fibroids. From these, 1 case had a giant myoma of about 16 cm in a pregnancy obtained by IVF (in vitro fertilization), and 1 woman had seven myomas. The second best indication for those who had a CS was having one’s first delivery after 35 years of age (10 cases), because the presence of fibroids is associated with this age. The third place indication for those who had a CS was having a uterus with scars (11 cases) (only one case after a myomectomy and other cases after a CS). Breach presentation occurred in primiparous women (4 cases), and arrest of dilatation (1 case), acute fetal distress (1 case) and narrow pelvic outlet (1 case) were also reported.

One hysterectomy was performed at the time of CS in a 42-year-old patient with seven fibroids due to poor uterine contractility caused by a large number of fibroids, with significant bleeding. During this study, the rate of myomectomies during CS increased to 24.48% compared with previous years regarding fibroids located near uterine incisions (Figure 8, Figure 9 and Figure 10) and those which were subserous (Figure 11, Figure 12, Figure 13, Figure 14 and Figure 15), but also regarding intramural ones and rarely those with submucous locations (Figure 16). 

From the 49 cases with birth deliveries by CS, we had only 40 women with known antepartum and postpartum hemoglobin values. Thus, for 31 cases without a myomectomy, we had an average of 12.16 g% antepartum and 10.4 g% postpartum, with a difference of 1.76 g% (*p* = 0.001). For the remaining 9 cases with a myomectomy, we had average antepartum hemoglobin of 12.46 g% (limits 10.4 g%–12.6 g%) and a postpartum average of 9.5 g% (limits 8 g%–11.6 g%), with a difference of 2.95 g% (*p* = 0.001) (Figure 16). Comparing blood loss between the two groups of CS, with and without myomectomy, the difference is not statistically significant (↓2.96 g% vs. ↓1.78 g%, *p* = 0.121). During CS, we used procedures to reduce blood loss, such as prophylactic sublingual administration, intrarectal misoprostol administration, oxytocin infusion or intravenous carbetocin in bolus administration.

For some patients, we inserted an intraperitoneal drain for the early identification of intraperitoneal hemorrhages and for the prevention of intraperitoneal collection of blood, which was removed after 24–48 h. Hospitalization was comparable between the groups with CS with a myomectomy compared with those without a myomectomy (4 days), but operating time was doubled for the group with a myomectomy associated with CS. No hysterectomy was required in the group with a myomectomy. There was no postoperative fever in any of the two groups of patients. Moreover, none of the patients received a blood transfusion. 

## 4. Discussion

Pregnant patients with fibroids are exposed to a high rate of complications during antepartum, intrapartum, and postpartum periods. The prevalence of uterine fibroids during pregnancy reported in some studies ranges from 1.6 to 16.7%, varying from one trimester to another [15,16,17]. In our study the prevalence was only 0.75% of all births during a period of 2 years and 6 months.

Previous data show that the number of fibroids increases with the patient’s age [18]. Unlike the literature data, which indicate that fibroid distribution in pregnant women increases beyond the age of 35, the incidence of fibroids was highest in the 30–34-year age group in our study.

The presence of fibroids in very young women can be correlated with a strong family history [19], but in our group none of the women mentioned fibroids in their hereditary collateral history.

Adipose tissue is a recognized extra source of estrogen, which is thought to play a role in the development of fibroids. The prevalence of overweight and obesity is on the rise, being higher in urban area and among educated women [1]. However, in the present study, only 19.3% of patients were obese.

It seems that nulliparity plays an important role in the etiology of fibroids. It is known that circulating hormones, such as estrogen and progesterone, are considered modulators for tumor growth. Consequently, fibroids should develop more frequently in pregnancy. However, according to the recent data that have looked into it, most fibroids do not develop or decrease during pregnancy. In addition, the multiparity plays a protective role by remodeling the uterine tissues [20]. Within our study, the majority of cases involved women with their first delivery. 

In our research, diagnoses were made in the first trimester. This can be explained by the fact that more and more pregnant women are going to the gynecologist to evaluate pregnancies from the first trimester, which makes it easier to establish the diagnosis of associated fibroids, with the uterus and pregnancy being small. The detection and evaluation of fibroids as the pregnancy progresses is difficult due to changes in uterine anatomy and the presence of the fetus and placenta that sometimes make examination difficult. 

The main effect of pregnancy on fibroids is related to the size of the uterine fibroids. For decades, scientists have debated whether hormonal changes that occur during pregnancy can affect the sizes of uterine fibroids. Uterine fibroids were considered to enlarge during pregnancy for several decades, especially during the first trimester. Benaglia et al., in their prospective cohort study on 25 women with fibroids, reported that, during the first 7 weeks of pregnancy, the sizes of the fibroids grew significantly to more than double their initial sizes [21]. Contrary to the data in the literature which mention the highest increase in the first trimester under the influence of estrogen and HCG, in the present study, in many cases, the maximum increase occurred in the second trimester. The ultrasound evaluation was done at the first visit at doctor in pregnancy, then at the morphology of trimester I, i.e., 12–13 weeks, at 28 weeks and at birth. The size considered in the growth assessment was the maximum one and only of the largest fibroid when there were more than one.

On the other hand, some studied have reported that up to 78% of uterine fibroids do not show significant growth during pregnancy [22]. We also had 10 cases in which fibroids remained constant in size during pregnancy. 

Placental abnormalities may also arise in pregnancy when uterine fibroids co-exist. Recent data have shown a 3-fold increase in occurrences of abruption placentae in pregnant women with uterine fibroids, particularly if the tumors have a volume of more than 200 cm^3^ and if they have a retroplacental or submucous location [9,10]. We had only 1 case with abruptio placentae. Epidemiological reports have noted a 2-fold increase in the risk of having *placenta praevia* in pregnant women with uterine fibroids [9,10].

A large fibroid can be problematic because it contains vessels, which can take over the blood needed for the vascularization of the uterus and fetal development. We had 3 cases with an absence of fetal cardiac activity in the first trimester, possibly caused by large fibroids that had sizes of 8–10 cm. Other data show that uterine fibroids do not tend to limit fetal growth during pregnancy; nevertheless, giant fibroids may cause fetal malformations, such as limb reduction deficits, dolichocephaly and torticollis [8,23].

Labor and delivery are more difficult in the case of fetal malpresentations and malpositions, which can be a cause of uterine distortions produced by fibroids. In addition, they raise the risk of complications for both the mother and the baby as well as the likelihood of operational interventions via cesarean section. Fibroids developing in the lower uterine region, with sizes of >5 cm, and being numerous are both risk factors for malpresentation [9,24,25]. In the study, there were 4 cases with breach presentation.

Previous data have shown that pregnant women with uterine fibroids had a 2-fold incidence in labor dystocia [10,26]. Women with uterine fibroids are more likely to experience third-stage labor difficulties and puerperal problems, and they are more likely to have a retained placenta. In addition, they tend to have dysfunctional labors as a consequence of the interference of myometrial contractility, and as a result they are more likely to utilize oxytocics to control uterine contractions and to ensure that the labor advances properly. After the baby and placenta are delivered, this effect on myometrial contractility repeats spontaneously, resulting in uterine atony and postpartum hemorrhaging [25]. As a consequence, the use of oxytocics, mechanical placenta removal, interventions to prevent postpartum hemorrhaging and using antibiotics peripartum are much more common in parturients with uterine fibroids [27]. Regarding the administration of antibiotics, our protocol stipulates intraoperative antibioprophylaxis with 1.5 g Cefuroxim, if the pregnant woman has intact membranes. Only 4 women had normal vaginal delivery in the present study, but we used oxitocin properly during labor and in the first 4 h postpartum. 

Pregnancy has a deep influence on uterine fibroids, including an increase in fibroids’ volume in 20–30% of cases, infection, torsion of pedunculated uterine fibroids, expulsion, red degeneration and necrosis. Fibroid pain as a result of red degeneration is a common concern, and it is normally treated conservatively with fluids, analgesics and hospitalization. As a result of the risk of fetal and neonatal adverse effects, such as intracranial hemorrhaging, premature closure of the fetal ductus arteriosus, necrotizing enterocolitis, pulmonary hypertension and oligohydramnios, non-steroidal anti-inflammatory drugs should be administered with prudence, especially in the third trimester [28]. We used non-steroidal anti-inflammatory drugs to relieve pain but only up to 30 weeks of gestation.

Myomectomies may merely extend the surgical duration, but they have a number of advantages, such as the possibility of avoiding a second procedure to remove fibroids [29]. For women of ages > 40 years, tendency to bleed, multiple myomas, and myomas located posteriorly or cornual are some of the indications of cesarean myomectomy. Blood transfusion, intraoperative bleeding, abnormal placental insertions and uterine rupture are some of the contraindications of cesarean myomectomy [29]. Myomectomies during CS doubled the operative time in our study. Pergialiotis et al. found in their study that women who had a myomectomy during cesarean delivery had a slight decrease in hemoglobin (mean difference 0.25 mg/dL, 95% CI 0.06–0.45), a relatively long surgical time (mean difference 13.87 min, 95% CI 4.78–22.95) and a higher number of postoperative hospital stays days (mean difference 0.35 days, 95% CI 0.25–0.46). Blood transfusion (odds ratio [OR] 1.41, 95% CI 0.96–2.07) and postoperative fever (OR 1.12, 95% CI 0.80–1.56) rates did not differ across groups [30]. In our study, patients with myomectomies had no significant difference in their length of postoperative hospitalization, rates of blood transfusions, and postoperative fever. A study by Goyal M et al. found no difference in incidences of hemorrhaging [RR = 1.16, 95%CI = 0.86–1.56, *p* = 0.32; moderate quality evidence] and fever RR = 1.17 [31], similar to our results.

As a result of the known potential of significant hemorrhaging, it is commonly required that myomectomies be avoided as much as possible during cesarean delivery. Nevertheless, there are numerous data related to successful myomectomies during cesarean section [26,28,32]. As a result of the higher risk of intrapartum uterine rupturing, women who already have myomectomies antenatally are more likely to have a cesarean section. Despite this, some of them have successfully vaginally delivered [29,33]. When a myomectomy is used to treat fibroids, there is a high risk of recurrence. Due to the development of additional fibroids, 10% to 25% of patients who choose a myomectomy require a repeated myomectomy in the future. Recurrences of uterine fibroids have demanded recurrent myomectomies, thus increasing the risk of complications in such patients [28].

A patient’s frequent request is the simultaneous surgical removal of a previously diagnosed myoma during CS. The myomectomy decision was considered difficult due to bleeding, because the pregnant uterus has increased vascularity; however, the recent literature suggests that this may be an option. For this reason, we analyzed comparative blood loss between cases with and without myomectomies using hemoglobin as an assessment parameter before and 2 days after CS. As stated before, we used procedures to minimize blood loss, including 200 mg misoprostol sublingually and intrarectally if necessary, oxytocin infusion or intravenous administration of carbetocin in bolus during CS. The amount of blood lost correlates with the size of the fibroid and with the subserous or intramural location. The sizes of the excised fibroids in our study ranged from 40 to 140 mm, with an average of 60 mm, most of which were subserous. For multiparous patients, bleeding was higher at birth due to uterine hypotonia. In our study, only one was tertiparous, one was at the second birth and the rest were nulliparous.

Because our study includes only 57 cases, it is difficult to assess the impact of fibroids on the course of pregnancy and to identify the specific features of fibroids that are truly important. The mechanisms by which fibroids cause obstetric adverse events are not clearly understood. The available data are difficult to interpret due to the heterogeneity of the study population, which differs in number, location (lower uterus or uterine fundus), type (submucosal, intramural or subserosal; pediculate or sessile) and gestational age at diagnosis. Although the majority of these cases are asymptomatic, some are prone to developing complications and may end up having adverse outcomes in pregnancy.

## 5. Conclusions

Because of the risk of excessive hemorrhaging, obstetricians usually avoid the removal of uterine fibroids during cesarean deliveries unless they are tiny and pedunculated. Despite the fact that the majority of fibroids are asymptomatic, their location and size may have an impact on the pregnancy and delivery process. In our study, the growth of fibroids was continuous, and only 5 cases showed a decrease in size in the third trimester compared to the second one. Performing routine myomectomies during cesarean section is not indicated, but it is a feasible and safe technique in some cases, with a good prognosis for the patient. Consequently, the decision of performing myomectomies during pregnancy can be a challenge and must be performed for select cases. This procedure may have several benefits, such as avoiding another operation to remove fibroids.

## Figures and Tables

**Figure 1 healthcare-10-00855-f001:**
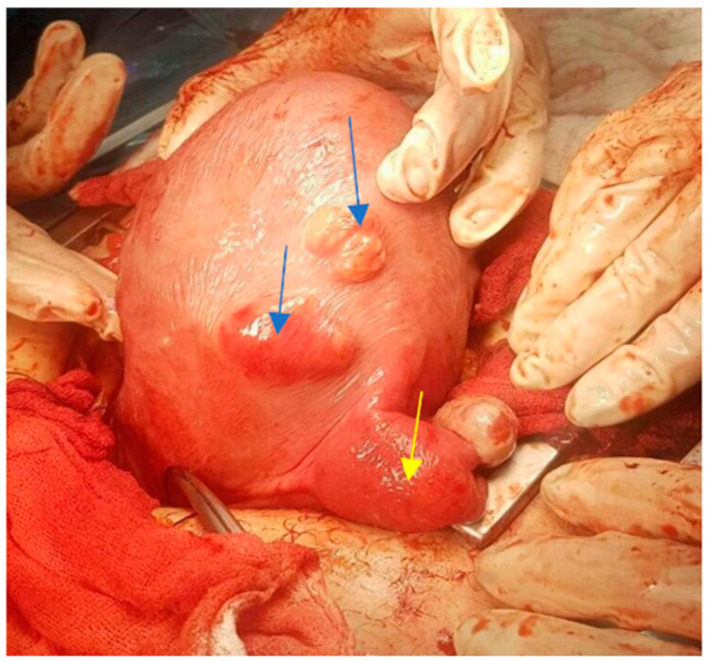
Fibroids of small dimensions found during CS (blue arrows), and diagnosis of double uterus (rudimentary left uterus) (yellow arrow).

**Figure 2 healthcare-10-00855-f002:**
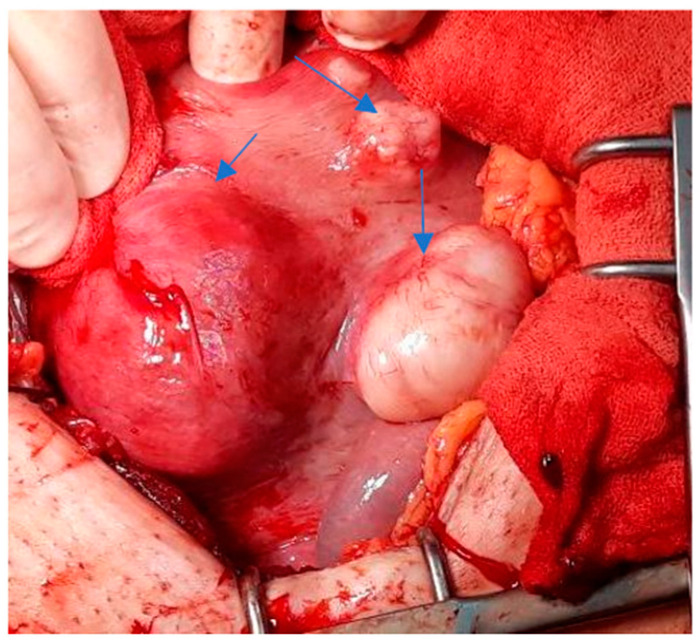
Uterus with multiple fibroids (blue arrows).

**Figure 3 healthcare-10-00855-f003:**
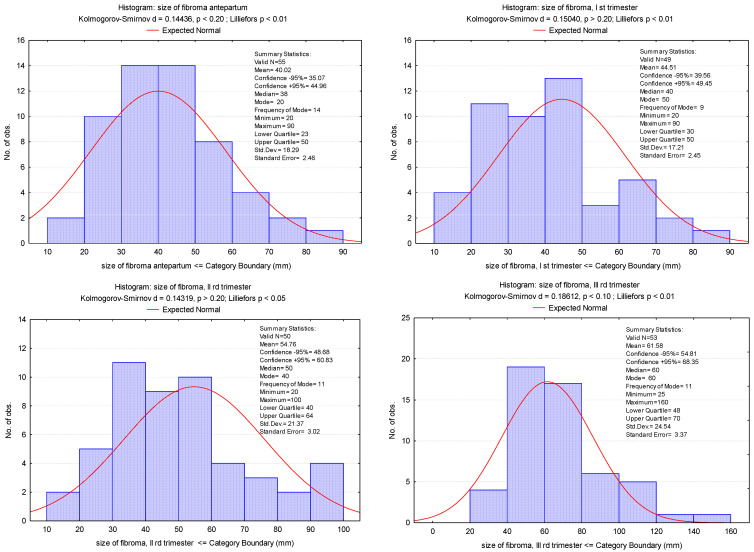
Fibroid dimensions antepartum and in each trimester of pregnancy.

**Figure 4 healthcare-10-00855-f004:**
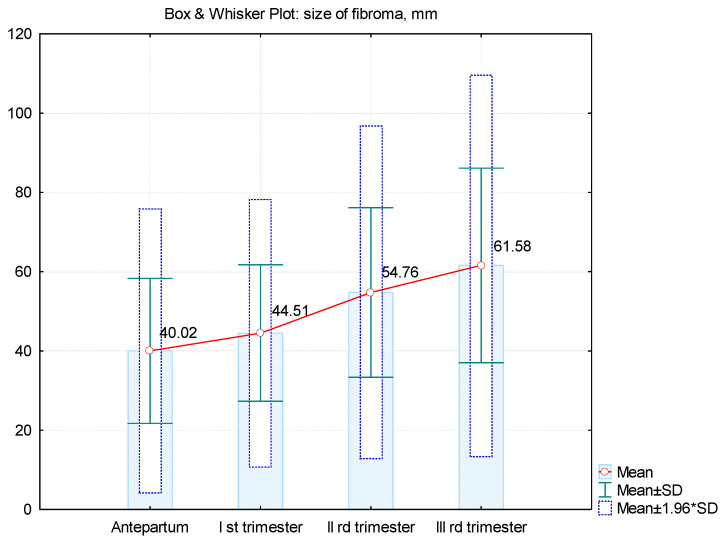
Mean values of dimensions of fibroids increasing during pregnancy.

**Figure 5 healthcare-10-00855-f005:**
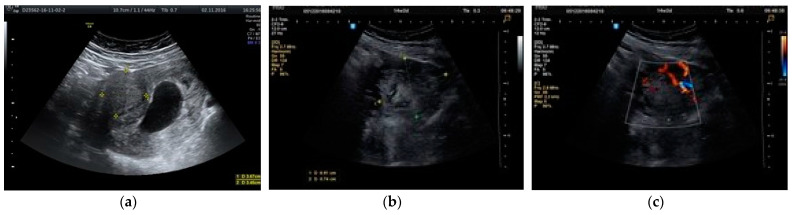
(**a**) Ultrasound scan at 9 weeks of pregnancy; fibroid of 36 mm. (**b**) Ultrasound scan at 14 weeks of pregnancy (same case); fibroid increased to 70 mm. (**c**) Doppler exam of the same fibroid.

**Figure 6 healthcare-10-00855-f006:**
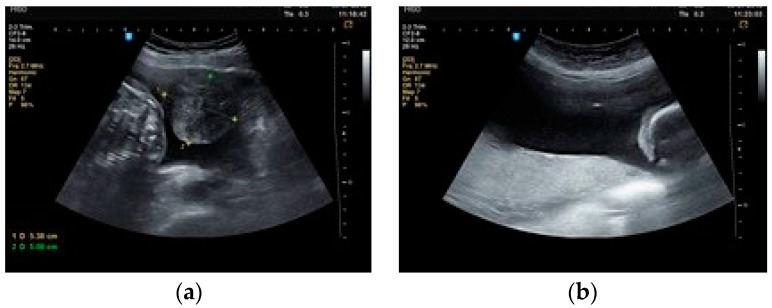
(**a**). Intracavitary fibroid praevia. (**b**). Needle inside the amniotic cavity in the superior part of the uterus.

**Figure 7 healthcare-10-00855-f007:**
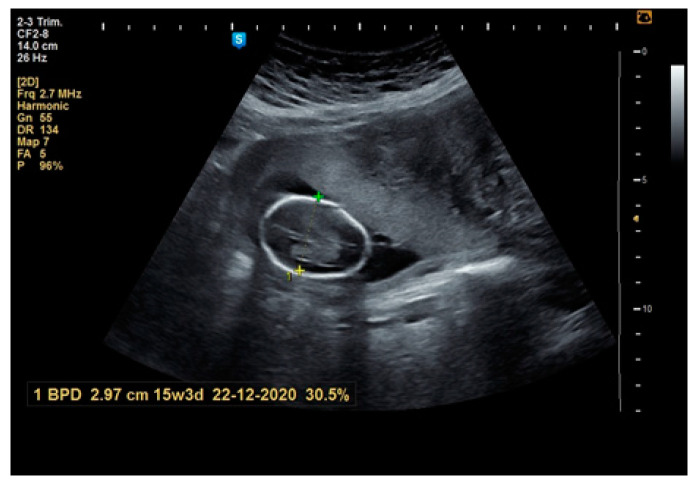
Anterior intramural fibroid and anterior placenta.

**Figure 8 healthcare-10-00855-f008:**
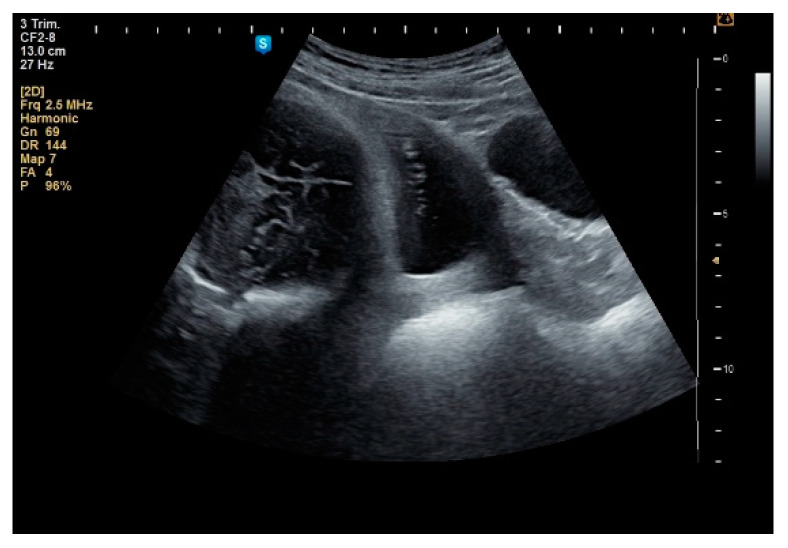
Ultrasound scan: pregnancy at term.

**Figure 9 healthcare-10-00855-f009:**
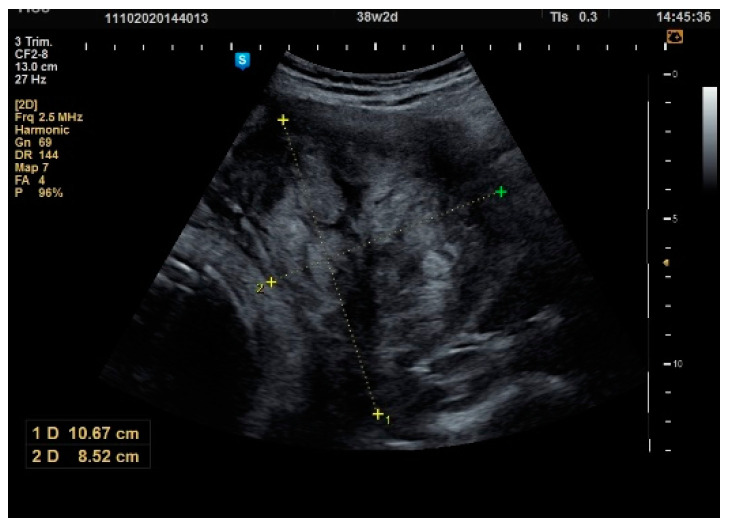
Cephalic presentation with intramural fibroid, lateral right, 143/100 mm.

**Figure 10 healthcare-10-00855-f010:**
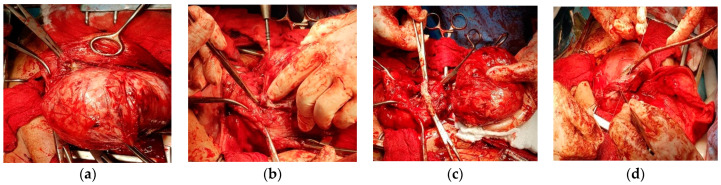
Aspects during CS: (**a**) After evacuation of the fetus, uterine contractility pushed the fibroid inside the opened uterine cavity; (**b**) The required myomectomy; (**c**) Aspect after excision of the fibroid; (**d**) Suture of the uterine transversal incision.

**Figure 11 healthcare-10-00855-f011:**
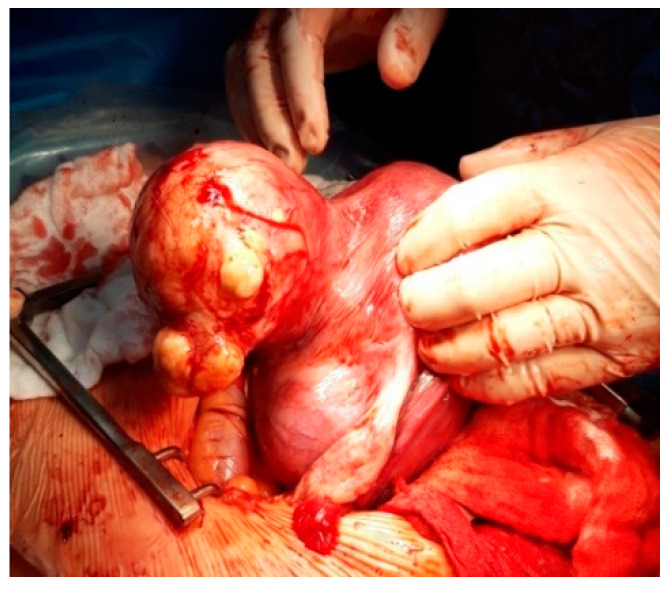
Subserous fibroid on pregnancy.

**Figure 12 healthcare-10-00855-f012:**
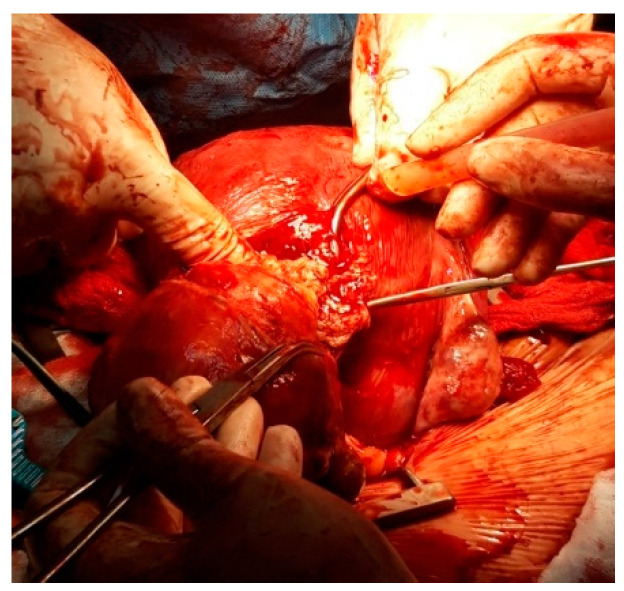
Myomectomy during CS.

**Figure 13 healthcare-10-00855-f013:**
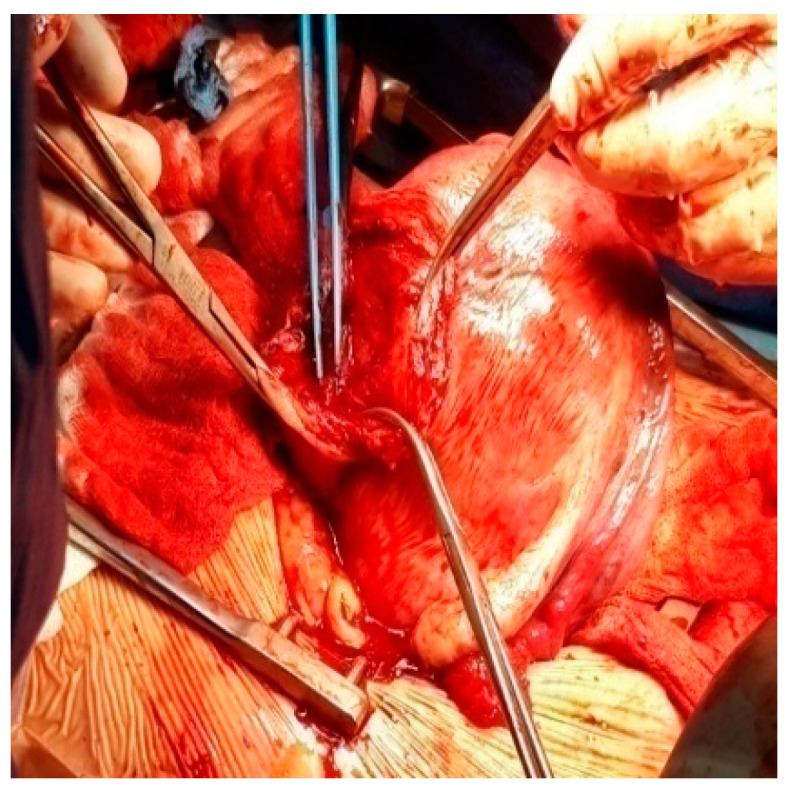
Uterus after myomectomy.

**Figure 14 healthcare-10-00855-f014:**
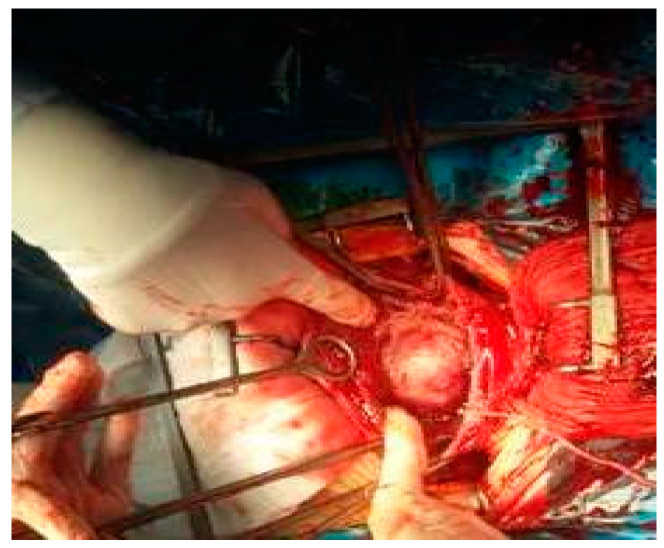
A 7 cm intracavitary fibroid praevia.

**Figure 15 healthcare-10-00855-f015:**
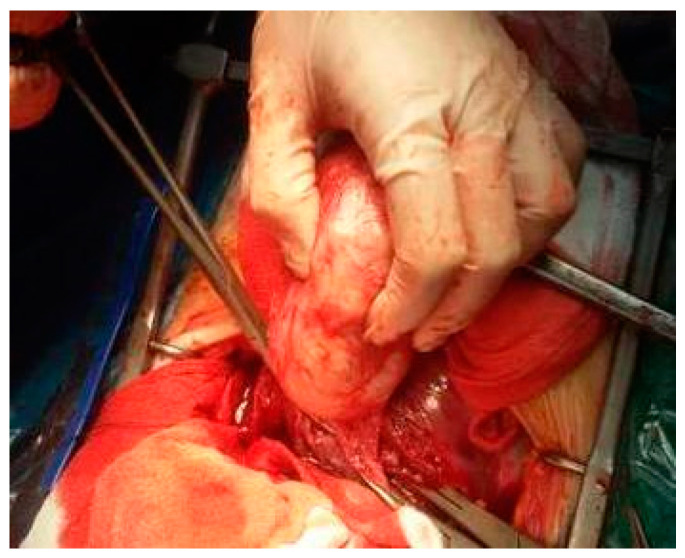
Myomectomy by sectioning the pedicle.

**Figure 16 healthcare-10-00855-f016:**
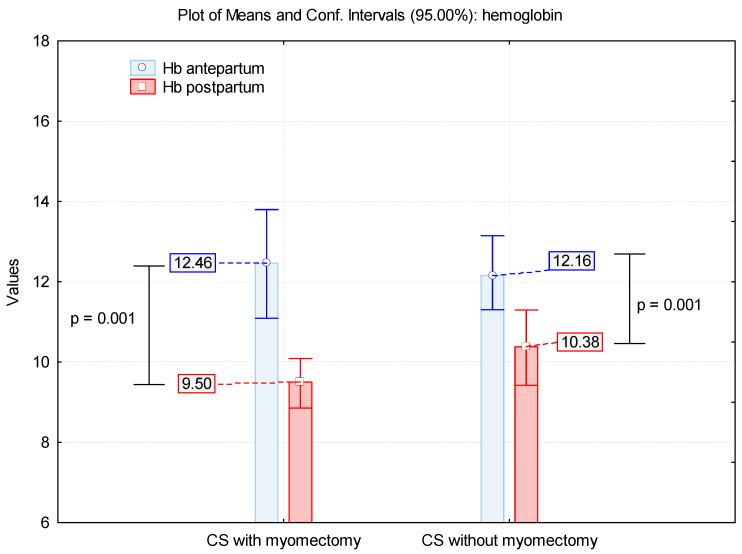
Mean values of hemoglobin in patients with CS with/without myomectomy.

**Table 1 healthcare-10-00855-t001:** Clinical and demographic characteristics.

Characteristics	Overall n = 57	CS with Myomectomy n = 12	CS without Myomectomy n = 37	Abortions n = 4	Normal Births n = 4	*p*-Value
**Age** in years, (mean ± SD) §	34.5 ± 4.2	34 ± 4.3	34.3 ± 3.7	38.3 ± 4.5	33.5 ± 3.7	0.4008
Age, median (range) **	35 (31; 37)	34 (30.5; 37.5)	34 (31; 37)	35 (31; 35)	39 (34; 42)	
**Environment**, U/R, n(%) †	46/11 (80.7/19.3)	10/2 (83.3/16.7)	31/6 (83.8/16.2)	3/1 (75/25)	2/2 (50/50)	0.4262
**Previous abortions**, n(%)						
without one abortion more than two abortions	40 (70.3) 7 (12.2) 10 (17.5)	7 (58.4) 1 (8.3) 4 (33.3)	27 (72) 6 (16.2) 4 (10.8)	4 (100) - -	2 (50) - 2 (50)	0.2111
**Obese**, Yes/No, n(%)	11/46 (19.3/80.7)	2/10 (16.7/83.3)	6/31 (16.2/83.8)	2/2 (50/50)	1/3 (25/75)	0.5198
**Previous births**						
without previous births single births multiple births	37 (64.9) 17 (29.8) 3 (5.3)	8 (66.7) 4 (33.3) -	26 70.3) 9 (24.3) 2 (5.4)	2 (50) 2 (50) -	1 (25) 2 (50) 1 (25)	0.4038
**Gestational age**, weeks						
mean ± SD median (range)	38.2 ± 1.7 39 (38; 39)	38.1 ± 1.2 38 (38; 39)	38.3 ±1.6 39 (38; 39)	-	37.3 ± 3.6 38 (35; 39)	0.7464
**Intrauterine Growth** **Restriction**, n(%)	6 (10.5)	2 (16.7)	4 (10.8)	-	-	0.7352
**Time of diagnosis with fibroids**						
before pregnancy during pregnancy	26 (45.6) 31 (54.4)	4 (33.3) 8 (66.7)	18 (48.6) 19 (51.4)	2 (50) 2 (50)	2 (50) 2 (50)	0.8184
**Number of fibroids**						
single fibroids multiple fibroids	42 (73.7) 15 (26.3)	8 (66.7) 4 (33.3)	27 (73) 10 (37)	3 (75) 1 (25)	4 (100) -	0.4324
**Type of fibroid**						
interstitial subserous submucous	-	28 (49.12) 26 (45.62) 3 (5.26)	-	-	-	
**Location of the myoma**						
anterior uterine wall fundic region posterior wall isthmic region right lateral walls left lateral walls	-	29 (50.87) 11 (19.3) 7 (12.28) 6 (10.52) 1(1.75) 3 (5.26)	-	-	-	

§ Kruskal–Wallis ANOVA by Ranks; Student’s *t*-test; † Pearson’s Chi-squared test; ** Values presented as median (range: Q25–Q75).

**Table 2 healthcare-10-00855-t002:** Fibroma dimension variation in all trimesters of pregnancy.

	Size of Fibroid, mm			
	Antepartum	1st Trimester	2nd Trimester	3rd Trimester
N	55	49	50	53
Mean	40.02	44.51 *^p^* ^= 0.005 *^	54.76 ^*p* = 0.001 *^	61.58 ^*p* = 0.001 *^
Difference from the previous average	-	+4.49 mm (+11.2%)	+10.25 mm (+23.0%)	+6.92 mm (+12.6%)
Median	38	40	50	60
Std. Deviation	18.29	17.21	21.37	24.54
Variance	42.88	38.67	39.04	39.59
Skewness	0.970	0.733	0.664	1.636
Std. Err. of Skewness	0.322	0.340	0.337	0.327
Minimum	20	20	20	25
Maximum	90	90	100	160
25th percentile	23	30	40	48
50th percentile	40	40	50	60
75th percentile	50	50	64	70

* *p* values for Student’s *t*-test between antepartum and trimesters of pregnancy.

## Data Availability

The data presented in this study are available on request from the corresponding authors. The data are not publicly available due to privacy.

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
