# Peer review of "Uterine Fibroids and Pregnancy: A Review of the Challenges from a Romanian Tertiary Level Institution"

_healthcare, 2022, doi:10.3390/healthcare10050855_

Round 1
Reviewer 1 Report
Thank you for your article on uterine fibroids in pregnancy.
Line 79- 80 would break this into 2 sentences - 1st on how you diagnose fibroids and second on time period.
Line 80 - this should be first part of result section with prevalence calculation
Line 82 - this is confusing - are only patient delivering by csection being included in study? If so I would change your title to reflect the question being studied (ie blood loss with myomectomy at time of c-section or characteristics of fibroids during pregnancy course?)
95 - Would start with prevalence of fibroids and incidence of c-section and myomectomy in pregnant patients
96 - this should be a table with demographics of women with fibroids in pregnancy.
Figures 1-3- do not think these pictures add much to the story. If you keep these images would classify by WHO category of fibroid and add measurements. A better picture would be of the submucosal fibroid with pregnancy as that is a known cause of miscarriage/infertility and would be more interesting than intramural
For graphs after line 130 would put mean, mode and range of number of ultrasounds in patients to clarify how many timepoints/usn there were to evaluate growth
The pyramidal growth figure is very difficult to visualize. I think a 2d bar graph would be better (or a circle with proportional increase by trimester?)
I would make paragraph starting with Line 146 into table
Figure 13b myomectomy (not miomectomy)
Figure 14 - don't know how helpful this figure is as uterine remodeling takes longer than 3d. Would be more interesting to scan with saline sonogram 4-6 wks out to check for presence of isthmocele
Line 180 - were there any cases of csection with myomectomy with large EBL? Outliers may be interesting (ex size, location, gestation, etc) Also, what types of fibroids were removed at time of c-section?
Line 197 - can you comment on intradeliver antibiotics used? Same for c-section with myomectomy?
Line 205 - would be interesting to note prevalence of assisted reproductive technology and infertility in this series
Line 238 - what is hypothesis for large growth in second trimester if not hormonal?
What were indications for c-section in these women?
Do you have a hypothesis with power calculation to see if your n is high enough to answer primary question?
Author Response
Dear reviewer,
Thank you for all your comments.
- Line 79- 80 would break this into 2 sentences - 1st on how you diagnose fibroids and second on time period.
Thank for your suggestion! We did that!
- Line 80 - this should be first part of result section with prevalence calculation!
Thank for your suggestion! We did that!
- Line 82 - this is confusing - are only patient delivering by csection being included in study? If so I would change your title to reflect the question being studied (ie blood loss with myomectomy at time of c-section or characteristics of fibroids during pregnancy course?)
All women diagnosed with pregnancy-related fibroids during that time period were included in the study in order to have a general picture of this association. As we reported in Material and methods, 49 of the 57 had caesarean sections, four had normal births, and four had abortions.
- 95 - Would start with prevalence of fibroids and incidence of c-section and myomectomy in pregnant patients.
We added the information also in a table (Table 1).
Our study aimed to analyze the cases in which there was a fibroid-pregnancy association and not the analysis of the incidence of fibroids / cesarean section in the total number of pregnant women. Thank you!
- 96 - this should be a table with demographics of women with fibroids in pregnancy.
We added the information in a table (Table 1). Thank you!
- Figures 1-3- do not think these pictures add much to the story. If you keep these images would classify by WHO category of fibroid and add measurements. A better picture would be of the submucosal fibroid with pregnancy as that is a known cause of miscarriage/infertility and would be more interesting than intramural
We removed pictures 1-3. Thank you!
- For graphs after line 130 would put mean, mode and range of number of ultrasounds in patients to clarify how many timepoints/usn there were to evaluate growth
We added all data you requested! Thank you!
- The pyramidal growth figure is very difficult to visualize. I think a 2d bar graph would be better (or a circle with proportional increase by trimester?)
We changed the charter type! Thank yoy!
- I would make paragraph starting with Line 146 into table
We consider that a table is not necessary for the respective paragraph regarding the evolution. Thank you!
- Figure 14 - don't know how helpful this figure is as uterine remodeling takes longer than 3d. Would be more interesting to scan with saline sonogram 4-6 wks out to check for presence of isthmocele
We just wanted to show the postoperative aspect of the area where the fibroid was removed when the patient was discharged. But we removed the figure. Thank you!
- Line 180 - were there any cases of csection with myomectomy with large EBL?
We did not have cases with EBL because we carefully selected the cases in which myomectomy was decided according to indications and contraindications for myomectomy during cesarean section. With this regard we added a paragraph to the Discussions section. Thank you!
- Outliers may be interesting (ex size, location, gestation, etc) Also, what types of fibroids were removed at time of c-section?
The amount of blood lost correlates with the size of the fibroid and the subserous or intramural location. The size of the excised fibroids in our study ranged from 40 to 140mm with an average of 60 mm, most of which were subserous. At multiparous, the bleeding is higher at birth due to uterine hypotonia. In our study, only one was tertiparous (third birth), one at the second birth, and the rest nulliparous, with no other previous births. We added the information in the article. Thank you!
- Line 197 - can you comment on intradeliver antibiotics used? Same for c-section with myomectomy?
Regarding the administration of antibiotics, we use only intraoperative antibioprophylaxis regardless of myomectomy or not: Cefuroxim 1 dose of 1.5g if the pregnant woman has intact membranes. We added the information in the article. Thank you!
- Line 205 - would be interesting to note prevalence of assisted reproductive technology and infertility in this series
We aimed to talk about fibroids associated with pregnancy and not about infertility and the difficulty of getting pregnant in patients with fibroids. That is not the purpose of our paper. Anyway, we had in the database 3 IVF of which one got pregnant at the third fertilization attempt.
- Line 238 - what is hypothesis for large growth in second trimester if not hormonal?
We did not consider any hypothesis other than hormonal, but according to literature data the increase would be higher in the first trimester due to increased HCG associated with estrogen. From the second trimester it decreases and yet in the analyzed group the maximum increase was in the second trimester only under the action of estrogen. Maybe the number of cases could explain our results.
- What were indications for c-section in these women?
Caesarean section indications are already mentioned in the text. In addition we added a paragraph in the Discussion section with general indication and contraindication!
In our study 4 cases (7%) with vaginal delivery and 49 by CS (85.96%). Regarding the indication for cesarean section, 21 (42.86%) cases presented praevia fibroids. From these, 1 case was with giant myoma about 16 cm in a pregnancy obtained by IVF (in vitro fertilization) and 1 woman had 7 myomas. The second indication for CS was the first delivery after 35 years (10 cases), because the presence of fibroids is associated with this age. On the third place as indication for CS was uterus with scar (11 cases) (only one case after myomectomy and others after previous CS). Also, breach presentation in primiparous (4 cases), arrest of dilatation - (1 case), acute fetal distress (1 case), and narrow pelvic outlet (1 case).
- Do you have a hypothesis with power calculation to see if your nb (natural birth )is high enough to answer primary question?
Unfortunately, in Romania, many women with a history of c-section delivery give birth later, also by c-section. Very few give birth naturally on the uterus with a post-c-section scar.
Hope we have touched all the points you asked us to change.
If there are any other changes you consider we should make, please let us know.
Yours sincerely,
All the authors

Reviewer 2 Report
The paper needs 1 more in depth explanation of factors which are r to responsible for leiomyoma size changes during pregnancy,2answer the question about indications and contraindications to myomectomy during cesarean section ,3 presentation of clinical characteristic of group which have myomectomy and group without myomectomy,4 to answer the question: who decided about myomectomy : obsterician/ surgeon, patient? Is it described in PROSPECTIVE study protocol and informed patient consent
Author Response
Dear reviewer,
Thank you for all your comments.
The paper needs
- more in depth explanation of factors which are r to responsible for leiomyoma size changes during pregnancy
We did not research the factors that influenced the growth of fibroids in pregnancy. According to literature data, the increase would be higher in the first trimester due to increased HCG associated with estrogen. From the second trimester it decreases. However, the results of our study showed the maximum increase in the second trimester only under the action of estrogen. This could be the consequence of a low number of cases!
- 2 answer the question about indications and contraindications to myomectomy during cesarean section
We mentioned that the myomectomy was performed mainly for large subserous fibroids or for those located in the vicinity of the cesarean section in order to be able to suture the uterus.
In our study 4 cases (7%) with vaginal delivery and 49 by CS (85.96%). Regarding the indication for cesarean section, 21 (42.86%) cases presented praevia fibroids. From these, 1 case was with giant myoma about 16 cm in a pregnancy obtained by IVF (in vitro fertilization) and 1 woman had 7 myomas. The second indication for CS was the first delivery after 35 years (10 cases), because the presence of fibroids is associated with this age. On the third place as indication for CS was uterus with scar (11 cases) (only one case after myomectomy and others after previous CS). Also, breach presentation in primiparous (4 cases), arrest of dilatation - (1 case), acute fetal distress (1 case), and narrow pelvic outlet (1 case).
In addition, we added more information in Discussion section. Thank you!
- 3 presentation of clinical characteristic of group which have myomectomy and group without myomectomy
With this regard we addded a table (Table 1) with data you requested! Thank you!
- 4 to answer the question: who decided about myomectomy: obstetrician/surgeon, patient? Is it described in PROSPECTIVE study protocol and informed patient consent
Of course, the patient's informed agreement is required for this prospective study. The obstetrician makes the decision after discussing the possibility of a myomectomy with the patient in advance, whether it is convenient and technically feasible intraoperatively without causing an excessive blood loss or requiring a hysterectomy.
Hope we have touched all the points you asked us to change.
If there are any other changes you consider we should make, please let us know.
Yours sincerely,
All the authors

Round 2
Reviewer 1 Report
Good revision of title and objective. I would add low prevalence (57/7600) as percentage in line 97. Good use of edits.
